# Growth Performance and Feed Intake Assessment of Italian Holstein Calves Fed a Hay-Based Total Mixed Ration: Preliminary Steps towards a Prediction Model

**DOI:** 10.3390/vetsci10090554

**Published:** 2023-09-03

**Authors:** Damiano Cavallini, Federica Raspa, Giovanna Marliani, Eleonora Nannoni, Giovanna Martelli, Luca Sardi, Emanuela Valle, Marta Pollesel, Marco Tassinari, Giovanni Buonaiuto

**Affiliations:** 1Department of Veterinary Sciences, University of Bologna, Ozzano dell’Emilia, 40064 Bologna, Italy; damiano.cavallini@unibo.it (D.C.); giovanna.martelli@unibo.it (G.M.); giovanna.marliani2@unibo.it (G.M.); luca.sardi@unibo.it (L.S.); polleselmarta.dvm@gmail.com (M.P.); m.tassinari.unibo@gmail.com (M.T.); giovanni.buonaiuto@unibo.it (G.B.); 2Department of Veterinary Sciences, University of Turin, 10095 Grugliasco, Italy; federica.raspa@unito.it (F.R.); emaluela.valle@unito.it (E.V.)

**Keywords:** heifers, dairy calves, replacement herd, feeding management, calf welfare

## Abstract

**Simple Summary:**

Calves, and particularly heifers, play a crucial role in the future of any herd, making them an essential component of dairy farms. Rearing calves, especially those intended for replacement, involves a significant investment of resources and time to achieve profitability. Therefore, ensuring the optimal health and performance of calves is of the utmost importance on dairy farms. One effective approach is to provide calves with a hay-based ration, which promotes better intakes, reduces the sorting of forage and concentrates, and ultimately enhances their overall health and welfare. This study evaluates the growth parameters of calves fed on a hay-based ration, and it also proposes a preliminary prediction model that could help farmers to make informed management decisions and select the best animals for the replacement of culled cows, thereby improving the overall productivity and efficiency of the herd.

**Abstract:**

The aim of this study was to evaluate the effects of a complete hay-based total mixed ration (TMR) for calves, focusing on their feed intake, animal growth performance, and fecal output, and to develop a preliminary estimation equation for solid feed intake and body weight in Holstein heifer calves. Twenty female Italian Holstein calves (37.14 ± 2.72 kg) born between February and July were studied from the day of birth until 77 days of age. From the fourth day of life, they were fed 3 L/day of pasteurized milk twice daily and supplemented with the same hay-based TMR. The data on feed intake, fecal characteristics, and growth performances were collected and showed that these calves had adequate parameters. Moreover, the data collected was used to create equations to predict body weight and solid feed intake using a mixed model. The goodness of fit of the developed equations was evaluated by coefficients of determinations (R^2^). The equation obtained shows high R^2^ (0.98 for solid feed intake and 0.99 for calf weight), indicating the satisfactory precision and accuracy needed to predict female calves’ body weight and solid feed intake.

## 1. Introduction

Calves, and in particular heifers, are the future of any herd, which is why they are a key part of the dairy farm [1]. Rearing calves, particularly for replacement, is an expensive investment that takes a significant time to repay. Dairy farm management and, in particular, feeding practices strongly affect the welfare of animals, especially in the early stages of life [2]. Indeed, from birth to post-weaning, calves undergo great physiological and metabolic changes [3]. The weaning period, in particular, represents the initial significant feeding transition for calves, posing challenges for both the animal and the producer [4]. Consequently, the early introduction of forage, especially high-quality forage like alfalfa hay, plays a crucial role in alleviating weaning stress [5]. Considering this, the utilization of hay-based Total Mixed Rations (TMR) could be a favorable option for calf rearing due to its potential to optimize nutrient intake and facilitate early rumen maturation, a crucial factor for long-term productivity in dairy cattle. According to Gabler et al. [6], raising the replacement herd accounts for more than 12% of total farm expenses, with feed accounting for more than 60% of costs. The appropriate growth of young dairy heifers designated for replacement is crucial for a dairy farm [7]. In fact, the total expenses incurred for the rearing of replacement heifers represent approximately 25% of the total cost of milk production, making it the second largest expense after feed costs for a dairy farm [8]. The use of decision-making systems based on prediction models [9] could support farmers to enhance rearing conditions through a reduction in specific direct costs. De Marchi et al. [10] reported that mathematical models are increasingly being used in the dairy cattle sector for a wide range of predictions (from milk production to health). This approach is particularly useful for those categories, such as calves or heifers [11], where growth performance is closely linked to factors such as productivity, reproduction, and health [12]. Whichever modeling is used, it assumes the collection of an adequate amount of data relating to the category of animals, carried out over a significant period and under defined farming conditions. As far as dairy calves are concerned, the literature contains numerous studies on growth performance. However, most published prediction models are based on high-concentrate diets [11] or silages [13] or specific stages (e.g., pre-weaning) [14] or particular conditions (e.g., during transport) [15], and to the best of our knowledge, no studies on TMR diets for calves are available. Regression analysis has the potential to empower farmers with predictive tools to make informed decisions regarding the best productive destination for their calves, facilitating the decision-making process between rearing calves as replacement heifers or for sale.

Within this context, the present trial was designed with a dual aim: Firstly, to expand our knowledge on the effects of offering a complete TMR ration to healthy Holstein calves from one week of age up to two weeks after weaning on feed intake, growth, and health. Secondly, we aimed to develop an estimation equation to predict the calves solid feed intake and body weight under this feeding regime.

The rationale for employing regression analysis in this study lies in its ability to provide a comprehensive understanding of how distinct nutritional elements within TMR contribute to calf growth, health, and overall development and future performance. In summary, our investigation aims to bridge the gap between the theoretical knowledge of TMR feeding in calves and its on-farm application, equipping farmers with valuable information to make informed decisions that contribute to the overall well-being and productivity of their dairy herds.

## 2. Materials and Methods

### 2.1. Farm and Management

The experiment was carried out in a commercial dairy herd located in Northern Italy. All experimental procedures were carried out in compliance with Directive 2010/63/EU [16], and the animals were raised according to Directive 2008/119/EC [17]. The trial was authorized by the Ethical Committee of the University of Bologna, Authorization Protocol n° 1039. The calves were housed in the same barn and reared in individual hutches (2.2 × 1.5 m); the calves were able to have direct visual and tactile contact and had straw bedding. New straw was added weekly. The animals were checked daily by trained personnel to verify the absence of symptoms of disease. All calves received, using a bucket with a soft rubber nipple, four colostrum meals, 2 L/meal, in the first two days of life; in the morning of the third day, at 07:15 am, they received one pectin replacement meal, and in the afternoon, at 05:15 pm, they received a half pectin–half milk meal. From the fourth day of life, they were fed 3 L/day of pasteurized milk twice daily and supplemented with the same TMR ad libitum (Table 1). Pasteurized milk (Table 2) was provided once a day from 56 to 63 d of age, when the calves were weaned. Each calf had free access to an ad libitum amount of clean drinking water in a plastic bucket throughout the study. The quality of the hay was checked according to Cavallini et al. [18] to ensure the absence of molds and spores. The corn used had an aflatoxin level below the EU’s maximum tolerable threshold, assessed according to Girolami et al. [19]. According to farm protocols, all of the calves in our study were vaccinated. The calves received a dose of modified live vaccine for viral diseases (Parainfluenza-3 virus and respiratory syncytial virus) within the first week of life and a dose of killed vaccine for Bovine Parainfluenza-3 virus, Bovine Respiratory Syncytial Virus, and *Mannheimia haemolytica* at 28 d of age. Health checks, including fecal consistency and pH, as described below, were completed twice a week.

### 2.2. Experiment: Measurements and Sampling

Twenty female Italian–Holstein calves, 37.14 ± 2.72 kg, born between February and July 2019 were studied from the day of birth until their 77th day of life in order to cover the whole weaning period. All selected calves were born from multiparous Italian Holstein dams. At the start of the trial, all of the calves were in a good, healthy condition. A trained breed expert visually scored the BCS (Body Condition Score) of each calf using a 5-point scale [20].

All calves were individually weighed on a digital scale at birth during the pre-weaning period until the 42nd day of the study (at weaning), from the 42nd to the 63rd day, and on the 77th days (i.e., post weaning). Weighing was always carried out at the same time of the day, 1–2 h after their morning meal. Individual feed intake was recorded daily throughout the study by weighing the amount of feed, pasteurized milk, and TMR offered and the amount of pasteurized milk and/or TMR refused. A representative sample of TMR (500 g) was collected weekly immediately after preparation and frozen to preserve its chemical characteristics prior to subsequent analysis. The total amount of feed offered was adjusted daily to ensure 5 to 10% orts. Weight gain, feed intake, feed conversion ratio (FCR), and average daily gain (ADG) were calculated for each period (pre-weaning, weaning, post-weaning) and across the entire study. Individual fecal samples were obtained via manual stimulation of the rectal ampulla before feeding. Fecal pH was measured using a portable pH meter (model 250A, Orion Research, Boston, MA, USA). Fecal color and consistency were visually evaluated by a trained observer to evaluate the presence of diarrhea using a 2-point scale (for consistency)—0 = soft (does not hold form, piles but spreads slightly); 1 = firm (firm but not hard)—and the following 3-point scale for color: 1 = white; 2 = brown; 3 = green. After the measurements were taken, the fecal samples were immediately frozen pending chemical analyses.

### 2.3. Chemical Analysis

The pasteurized milk was sampled weekly from the farm pasteurizer and immediately frozen. All the milk samples were collected, pooled, and analyzed to determine fat, protein, and lactose concentrations (Table 2). TMR and feces samples were sent to the laboratory of the Animal Production and Food Safety service of the Department of Veterinary Medical Sciences (DIMEVET), University of Bologna, for dry matter (DM) and chemical analysis. To determine the DM content, the samples were dried in a forced air oven at 65 °C until a constant weight was achieved. Upon drying, the samples were ground to pass through a 1 mm screen (Cyclotec Mill, model 1093; Foss Tecator, Höganäs, Sweden). The ground samples were analyzed for ashes after 4 h of combustion in a muffle furnace at 550 °C (Vulcan 3–550, Dentsply Ney-tech, Burlington, NJ, USA) ash-corrected α-amylase–treated neutral detergent fiber (NDF) with the addition of sodium sulfite (aNDFom), acid detergent fiber (ADF), and acid detergent lignin (ADL) [21]; Crude protein (CP) using a Kjeldahl nitrogen analyzer (Gerhadt Vapodest 50, Gerhardt GmbH, Königswinter, Germany); Starch [22], method 996.11; and Ether extract (EE; according to EC Regulation No. 152/2009).

### 2.4. Statistical Analysis

All data were analyzed using Microsoft Excel and JMP Pro v 14.3 software. Numerical data were analyzed using a repeated measures mixed model procedure. Firstly, the data were tested for normality and homoscedasticity by using the Shapiro–Wilk and Levene tests, respectively [23]. Feed intake and live weight curves were developed according to the method used in a similar study [20] using the following polynomial regression model:$$y=day_{j}+\overset{n}{\underset{i=1}{\sum }}\beta_{n}P_{n}+\overset{n}{\underset{i=1}{\sum }}\alpha_{n}P_{n}+e_{jklm,}$$
where $y_{jklm}\;$ is feed intake or live weight measured during the study for the k* animal, $day_{j}$ is the fixed effect of the *j*-th time at measuring, $\beta_{n}$ is the *n*-th fixed regression coefficient of the polynomial modeling all records of feed intake or live weight throughout the experiment, $\alpha_{nk}$ is the *n*-th random regression coefficient of the polynomial modeling records of feed intake or live weight throughout the experiment for calf *k*-th, and $e_{jklm}$ is the random residual term.

For qualitative parameters, the chi-square test was applied. The significance level was set at *p* < 0.05 and is reported in the results as different superscripts.

## 3. Results and Discussion

The first aim of the present study was to investigate the effects of a complete TMR ration for healthy calves focusing on their growth performances, intake, and fecal characteristics. All of the parameters considered in the present investigation have a direct effect on calves’ intake, growing, health, and welfare.

### 3.1. Feed Intake

Regarding milk intake, no differences were found among the calves, who all finished their meals without leaving any waste. Additionally, milk intake observed at different timepoints was consistent with that observed in other studies [24]. As expected, TMR intake (TMRI, Table 3) increased as the animals grew older and the weaning time approached (Figure 1). Indeed, during the first days of the trial, the average value was very low (Figure 1), and only after 27 days did it reach and exceed 100 g. After this period, TMRI gradually increased as weaning approached (Figure 1). A similar trend was observed by Welk et al. [25] in calves fed with forage-based TMR. Some authors [26] have observed that early TMR administration to young livestock results in better ration consumption, reducing sorting against forage and for concentrates. According to Groen et al. [27], feeding heifers with a TMR from a young age not only has immediate behavioral effects but may also have longer-term benefits. For example, Xiao et al. [28] reported that early-life feed experience may affect the development of feeding behavior in female Chinese Holstein calves. Moreover, their study reported that early exposure to various feed sources and presentations may affect calves’ preferences, basically solving feed sorting problems.

Furthermore, calves, as lactating animals, have a digestive system similar to monogastric animals such as horses. As a result, a significant portion of the solid feed they consume bypasses the rumen. This characteristic has a limited effect, making it appropriate to exclusively provide calf feed. However, when concentrate feed is their sole source of sustenance, the hay is separated, leading to considerably lower fiber intake compared to more appealing feed options [29]. Our research suggests that incorporating a total mixed ration (TMR) could improve overall fiber intake compared to separate feeding methods. Therefore, we recommend implementing a slightly earlier weaning process and facilitating a gradual transition from monogastric to ruminant feeding.

Similarly, Miller-Cushon and DeVries [30] previously reported that calves exposed solely to concentrate tend to select short grain particles, while those exposed only to hay tend to select longer forage particles. Another example of this was reported by Greter et al. [26], who observed that heifers previously fed a top-dressed ration maintained similar feeding patterns (whereby they were slug-feeding their concentrate) after they were switched to a silage-based TMR for a period of 7 weeks.

### 3.2. Fecal Output

Table 4 shows the results related to the characteristics of the feces (pH and chemical analysis). Fecal pH is a crucial parameter for evaluating the digestive health of dairy calves since it can provide insights into the fermentation activity in the rumen and hindgut and allow for adjustments to be made to the diet accordingly. In the present study, fecal pH increased with animal growth from 5.40 on day 7 to 7.06 on day 42. This rise was justified by the increase in TMRI due to animal growth. Similar trends were observed by Kodithuwakku et al. [31] in female Holstein calves ranging from 7 to 49 days of age who were fed a solid diet from their first week of life. Therefore, monitoring fecal pH in dairy calves is an essential method for assessing their digestive health. Khorrami et al. [32] suggested that the fecal pH of dairy calves should be maintained within the physiological range of 6.0 to 7.0. In our study, values within this range were observed after the weaning period (6.23 at 63 days; 6.42 at 77 days). Regarding the color of the calves’ feces (Table 4), some variability among calves was observed only in the samples taken at 7 days of life, which may be due to differences in milk digestion capacity. However, all samples showed a brown coloration after day 7. Overall, there were no statistically significant differences in fecal color between the observations in younger and older calves (*p* = 0.35). No undigested feed particles were macroscopically visible during sample manipulation, processing, and analysis.

Regarding fecal consistency (Table 5), there was a clear prevalence of solid feces (71–78%) up to 63 days of life. At 77 days, solid feces were 56% and liquid feces were 44%. This may be due to the adaptation of the digestive tract to solid feed during weaning. However, there was no statistically significant difference in consistency between younger and older calves (*p* = 0.75).

In the present study, DM and CP content in feces (Table 4) decreased from 7 to 77 days of age (DM: from 20.1 to 14.0%; CP: 45.0 to 18.5%). Regarding fiber fractions, aNDFom, ADF, and ADL increased from the start to the end of the trial (e.g., aNDFom ranging from 11.2 at 7 days to 45.1 at 77 days). Fecal starch levels decreased during the same period from 2.3% to 1.1%. Lastly, the average ash content fluctuated, showing a minimum value of 6.51% at 63 days and a maximum value of 9.77% at 42 days of age. These results should be interpreted with caution, however, as the limited number of studies available on calves’ feces make it difficult to draw general conclusions.

### 3.3. Growth Performances

As depicted in Figure 2, the average birth weight of the calves that partook in the investigation was 37.14 ± 2.72 kg (range 95%: 34.82–39.46). This result is consistent with the scientific literature on specialized dairy breeds calves, including Holstein calves [33,34,35]. According to Renaud et al. [36], calves with a low body weight at birth have a greater risk of dying within 21 days and during the entire growing period [37,38]. Conversely, a higher body weight at birth has been associated with a lower incidence of respiratory disease and diarrhea in the two weeks following arrival on farm after transportation [39]. Rot et al. [40] also reported that greater body weight at birth was one of the most consistent predictors of future mortality and morbidity at calf-rearing facilities. In the present study, the average weight at 77 days (post-weaning period) was 84.72 ± 8.90 kg (Figure 2). Therefore, the average ADG during the trial was 0.618 ± 0.04 kg. The calf weight values in our study, which were measured at specific timepoints, were higher than those reported by Seifzadeh et al. [41] for replacement Holstein calves.

Looking across the different timepoints considered in the present study (Figure 2), the average weight gain in the first week was 3.50 ± 1.91 kg. During the pre-weaning periods (between 7 and 42 days of life), the average weight gain was 17.39 ± 3.67 kg; during the weaning phase (from 42 to 63 days of life), the average weight gain was 15.17 ± 3.74 kg, and in the post-weaning phase (from 63 to 77 days of life), the average weight gain was 11.52 ± 4.04 kg. The average ADG observed during the trial was 0.618 ± 0.125 kg. This result was similar to that observed by other researchers, such as that observed by Thomas et al. (0.61 kg/d) [42]. In the present study, ADG was 0.50 ± 0.27 kg during the first week, 0.49 ± 0.10 kg during the pre-weaning period, 0.49 ± 0.18 kg during the weaning phase, and 0.82 ± 0.29 kg during the post-weaning period. The ADGs observed are consistent with the range (from 0.3 to 1.3 kg/d) observed by Soberon et al. [43] for Holstein calves.

### 3.4. Preliminary Development of a Prediction Model: Practical Applications and Limitations

The data collected in the present investigation was plotted on a graph with a corresponding polynomial regression curve. The estimation equation was obtained using the method described in the statistical analysis section, which allows for the estimation of calf weight and feed intake at a specific age under similar conditions to those of the present study.

The equation obtained shows high R^2^ (0.98 for solid feed intake and 0.99 for calf weight; Table 6), indicating a good ability to predict solid feed intake and body weight. High accuracy (R^2^ > 0.9) of equations is the most important measure of goodness of fit and represents a model’s ability to predict actual values [44,45]. A similar result (R^2^ > 80) was reported by Guinguina et al. [46] for the prediction of feed intake and feed efficiency in lactating dairy cows using digesta marker techniques. Furthermore, the R^2^ values obtained in the present study were higher compared to those reported by Silva et al. [45], who reported a R^2^ value lower than 0.75 for the equation used to predict starter feed intake in pre-weaned dairy calves. It is important to highlight that the findings of Silva et al.’s study [45] revolve around a model crafted using data from numerous studies conducted across various farms. Their study specifically centered around a distinct feeding regimen that aimed to model the intake of starter feed. This distinction is significant, given that the composition of this starter feed diverges in terms of its compositional attributes from the feed regimen we employed in our investigation. Furthermore, it is worth noting that although Silva et al.’s study involved a larger number of animals, their primary focus was on male subjects. Additionally, their study encompassed animals of different genetic types (crossbreeds), meaning that it was solely concerned with Holstein breeds. This diversity in both gender and breed poses a challenge for making direct comparisons, as the characteristics of the subjects are not entirely analogous. This intricate interplay of varying factors underscores the complexity of drawing direct parallels between their findings and our study results.

The model’s predictive efficacy could represent a useful tool for farmers and on-farm advisors, particularly in its application to the management of upcoming heifers. For this reason, the equation developed in the present study could be readily integrated into a decision support tool to help dairy calf management support farmers to select the best heifers to include in future breeding stock. Indeed, the use of these equations can enhance farm management, improving the welfare of reared dairy calves by allowing for the early identification of discrepancies in feed intake or growth. 

Regarding the use of TMR in calves, feeding management during weaning and pre-weaning is essential to promote optimal growth, animal welfare, and future production [47,48]. Indeed, colostrum and milk administration are crucial for the growth and survival of calves [49]. However, the early supplementation of solid feed is crucial to stimulate the development of a functional rumen [50]. In addition, early intake of solid feed before weaning enhances calf growth, also increasing their chances of survival [24,51] and, consequently, their stayability [52]. Calf welfare may also benefit from this situation; in fact, calves consuming solid feed earlier are less likely to experience hunger in the first few days after weaning [53]. For this reason, usually, starter feeds have low fiber and high grain and legume contents to improve palatability [54]. However, a concentrate-rich diet is associated with some negative effects on calf health, such as impaired rumination and salivation, which in turn affect the digestive processes in the rumen and the overall development and health of the gastrointestinal tract [4]. Therefore, early forage provision (hays in particular) is accepted and largely recommended [55]. Calves with limited access to fiber may show rumen under-development, the presence of rumen plaques, and hyperkeratinisation, as well as abomasal lesions (ulcers), mainly in the pyloric region [56]. Moreover, hays could potentially be used to offset feed costs [57], although it should be remembered that calves have low ruminal capacity and particular nutrient requirements, so forage feeding should be carefully managed [58]. According to Groen et al. [27], feeding TMR provides a useful approach to promoting balanced nutrient intake within the day, also reducing feedbunk competition in animals raised in groups. Furthermore, TMR could be used to increase the use of feed produced on farm, contain feed costs, and maximize farm efficiency [59]. 

Overall, an interesting practical implication of this study is its relevance to the management of “surplus” calves within a farm, encompassing both economic considerations (animal potential and use of TMR) and welfare concerns (proper health and development of the future milking herd). In dairy herds, many calves are born every year without the need to replace the milking herd, and these are commonly referred to as surplus animals [60]. The majority of surplus calves are males [59]. However, as reported by Bolton and von Keyserlingk [60], the increasing utilization of sexed semen, combined with the growing demand for crossbreeding, has led to an increasing percentage of these excess calves being female. All these calves are sold at an early age for different purposes, including for veal or dairy–beef crossbred production [61]. As reported by Berry and Ring [62], certain dairy farmers tend to rear their surplus progeny after weaning. Selling these animals can provide a valuable cash injection during periods of financial strain. Using the data obtained in the present study, farmers could produce heavier (and consequently better-conformed) calves [63], resulting in heavier and higher-value primal carcass cuts if they are male or in heifers with better body conformation. These possibilities could be particularly interesting for heifer rearing, since studies [64,65,66] report a strong relationship between conformation traits and lifetime production efficiency.

We hope that this study can help towards the enhancement of farm management and dairy calf welfare through the early identification of feed intake or growth discrepancies. The early provision of solid feed supplementation aims to foster functional rumen development and improved growth and survival rates. The proposed equation can support decision-making, guiding both optimal heifer selection for future breeding stock and the management of surplus calves, potentially yielding economically beneficial outcomes. Overall, our findings show that administering hay-based TMR to Holstein Friesian heifer calves results in unaffected growth rates, feed intake, and fecal characteristics, providing valuable insights for calf rearing practices.

However, some limitations need to be acknowledged. First of all, it is obvious that the productivity of a dairy herd does not solely depend on calf management; instead, it is the result of a complex number of factors (e.g., culling policy, breed, lactation characteristics, milking management, veterinary input and herd health, farmer attitude, etc.). Therefore, our study specifically focused on calves’ projected potential. In addition, this preliminary model was developed on a single experimental group at one farm (female Italian Holstein calves); therefore, its generalizability remains to be assessed in multi-farm studies and meta-analyses and is yet to be externally validated. Recognizing these limitations is crucial to properly contextualize the study and the research questions that need to be investigated in order to gain a more comprehensive understanding of the impact of hay-based Total Mixed Rations (TMR) on calf development.

## 4. Conclusions

This study demonstrated that the administration of hay-based Total Mixed Rations (TMR) to Holstein Friesian heifer calves does not hinder their growth or adversely affect their feed intake and fecal characteristics. The growth rates of the calves remained within normal levels, and their feed intake and fecal characteristics were not adversely affected. These findings will help to fill a knowledge gap in the literature since there is a limited number of scientific research studies analyzing such parameters in healthy calves. Additionally, the equations in the present study showed high R^2^ values, indicating a good capability to predict solid feed intake and body weight. However, their applicability to independent data sets (including different farms, breeds, genders, etc.) needs to be evaluated. Hence, future studies should focus on external validation to assess the performance and reliability of the estimation equations across various calf populations.

## Figures and Tables

**Figure 1 vetsci-10-00554-f001:**
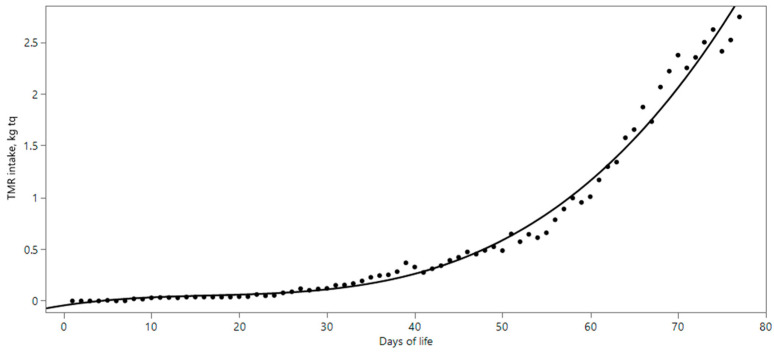
Average TMR Intake observed during the study.

**Figure 2 vetsci-10-00554-f002:**
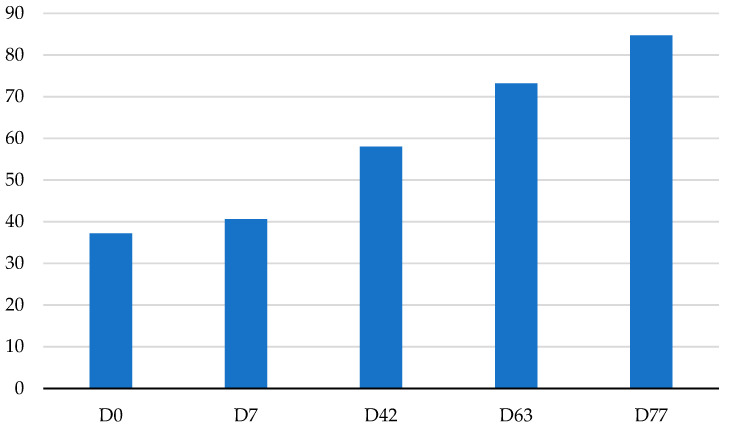
Average body weight of the calves involved in the present study (measured at different timepoints). D0: weight at the birth; D7: weight at 7 day; D42: weight at 42 days of life (pre-weaning period); D 63: weight at 63 days of life (finish of weaning period); D77: weight at 77 days of life (post-weaning period).

**Table 1 vetsci-10-00554-t001:** Ingredients (% of inclusion) and chemical composition (% of DM ± SD) of the hay-based total mixed ration (TMR) used in the study.

Ingredients	% of Inclusion
Mixture hay (1st cut)	18
Alfalfa hay	18
Corn flakes	36
Canola meal	13
Liquid feed (molasses blend)	13
Salt ^1^	1
Min-Vit supplement ^1^	1
Chemical composition	% of DM
CP ^2^	15.2 ± 0.9
Ether extract	1.7 ± 0.5
Ash	6.6 ± 0.4
aNDFom ^3^	28.3 ± 1.9
ADF ^4^	20.2 ± 1.5
ADL ^5^	5.1 ± 0.4
Starch	27.1 ± 2.7

^1^ Salt (20%), magnesium oxide (20%), urea (20%), bentonite (20%), stillage (15%), per kg: Vit A (1,500,000 UI), Vit D3 (800 UI), Vit E (2000 UI), Vit B1 (400 UI), Vit B2 (200 UI), Vit B6 (200 UI), Mn (10,000 mg), Zn (10,000 mg), Cu (1500 mg), Se (30 mg). ^2^ CP: crude protein; ^3^ aNDFom: amylase-treated ash-corrected NDF with the addition of sodium sulphite; ^4^ ADF: acid detergent fiber; ^5^ ADL: acid detergent lignin.

**Table 2 vetsci-10-00554-t002:** Chemical composition (% ± SD) of the pasteurized milk employed in the present study.

Milk Components	%
Fat	3.6 ± 0.2
Protein	3.2 ± 0.1
Lactose	4.8 ± 0.1

**Table 3 vetsci-10-00554-t003:** Growth performance recorded during the sub-periods of the trial.

Item	D7	D42	D63	D77	SEM
AWG, kg	3.50 ^D^	17.39 ^A^	15.17 ^B^	11.55 ^C^	0.84
TMRI, kg	-	4.4 ^C^	15.2 ^B^	29.9 ^A^	1.09
TMR/FCR	0.50 ^B^	0.49 ^B^	0.49 ^B^	0.82 ^A^	0.05

^A, B, C, D^ = *p* < 0.05 AWG: Average weight gain; ADG: Average daily gain; TMRI: Total mixed ration intake; TMR/FCR: Total mixed ration feed conversion ratio.

**Table 4 vetsci-10-00554-t004:** pH and chemical composition (% of DM) of fecal output.

Item	D7	D42	D63	D77	SEM
pH	5.4 ^C^	7.1 ^A^	6.2 ^B^	6.4 ^B^	0.1
DM	20.1 ^A^	15.5 ^B^	15.0 ^B^	14.0 ^C^	1.8
CP	45.0 ^A^	33.2 ^A,B^	20.5 ^A,B^	18.5 ^B^	5.2
Starch	2.3 ^A^	1.4 ^B^	1.6 ^B^	1.1 ^C^	0.6
aNDFom	11.2 ^C^	17.7 ^B^	48.8 ^A^	45.1 ^A^	3.9
ADF	9.6 ^C^	16.6 ^B^	39.5 ^A^	39.1 ^A^	10.2
ADL	3.4 ^C^	8.1 ^B^	13.4 ^A^	13.6 ^A^	3.9
Ash	7.1	9.8	6.5	8.4	4.4

^A, B, C^ = *p* < 0.05 DM: Dry matter; CP: Crude protein; aNDFom: a-amylase–treated NDF, ash-corrected; ADF: acid detergent fiber; ADL: acid detergent lignin.

**Table 5 vetsci-10-00554-t005:** Results from macroscopic fecal evaluations.

Item	D7	D42	D63	D77
Color				
White	14.3 ^A^	0 ^B^	0 ^B^	0 ^B^
Brown	71.4 ^B^	100 ^A^	100 ^A^	100 ^A^
Green	14.3 ^A^	0 ^B^	0 ^B^	0 ^B^
Consistency				
Soft	29 ^B^	25 ^B^	22 ^B^	44 ^A^
Firm	71 ^A^	75 ^A^	78 ^A^	56 ^B^

^A, B^ = *p* < 0.05.

**Table 6 vetsci-10-00554-t006:** Polynomial equations to predict solid feed intake and the weight of female Italian Holstein calves fed with hay-based TMR.

Equation ^1^	RMSE	R^2^
Solid Feed Intake		
Y_SFI_ = −0.571121 + 0.020813 × d + 0.0008355 × (d − 39)^2^ + 1.2525 – 5 × (d − 39)^3^	0.1	0.98
Body weight		
Y_W_ = 32.641424 + 0.6081959 × d + 0.0034102 × (d − 37.8)^2^	0.42	0.99

RMSE: Root Mean Square Error; R^2^: Coefficient of determination. ^1^ d= days of life.

## Data Availability

The data used in the current study are available from the corresponding author upon request.

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
