# Peer review of "Growth Performance and Feed Intake Assessment of Italian Holstein Calves Fed a Hay-Based Total Mixed Ration: Preliminary Steps towards a Prediction Model"

_vetsci, 2023, doi:10.3390/vetsci10090554_

Round 1
Reviewer 1 Report
The study presents preliminary results of the effects of a complete hay-based total mixed ration focusing on growth parameters and developing a preliminary estimation equation for solid feed intake and body weight in Holstein heifer calves.
Although the research is limited to a single experimental group in a specific farm setting, the results are well presented and discussed.
Moreover, it is well-written besides some minor typing errors that need to be corrected. The lines 346-357 need to be moved in the conclusion part.
Author Response
Reviewer 1:
The study presents preliminary results of the effects of a complete hay-based total mixed ration focusing on growth parameters and developing a preliminary estimation equation for solid feed intake and body weight in Holstein heifer calves.
Although the research is limited to a single experimental group in a specific farm setting, the results are well presented and discussed.
>>> AU: Thank you for appreciating our study.
Moreover, it is well-written besides some minor typing errors that need to be corrected. The lines 346-357 need to be moved in the conclusion part.
>>> AU: thank you, we double-checked the manuscript and corrected the typos. The sentence was moved as suggested (now lines 366-370)
Reviewer 2 Report
The title of this paper is, in simple mathematical terms, a polynomial regression equation where the dependant variable y may be either solid food intake or calf body weight, b the variable you can optimise, and x the independent variable. The authors include G. Buoniauto, and they expect the reader of this paper to go to reference 34 to find out the basic mathematics associated with a non-linear association between variables x and y, defined within that paper but relating to an entirely different age and breed of cattle than in the title of this paper. The justification for this paper is that it may help dairy farmers and their nutritionists/veterinary surgeons to manage better the nutritional requirements of calves during their first 70 days of life. Therefore, this paper should be written in a clear, simple way that describes the basic research methods used to provide the data that allowed this predictive model to be constructed, the variables defined within this model, the advantages and disadvantages of using this statistical method, and how it can be used to improve the health, welfare and future lactations of milking cattle so treated.
This objective has been lost completely in this paper: instead, it has become a review article of predictive models relating feed/rations with future performance/production of animals consuming grass/forages. Furthermore, 11% of the references come from the authors of this paper, and 18% of all references quoted are unrelated to the title of this paper. What this paper should be concerned with is the composition of the total mixed ration(TMR) fed to the dairy calves in this study, the overall nutritional management in the first 70 days of life, how this ensured that the regression equation derived was so consistent, and what that tells us about the physiological changes that occur in the rumen and digestive tract of calves within this crucial period of life which are primarily genet-related and can be influenced by external factors. These factors are important when relating your result to that of Silva - ref. 73.
There are several other general issues that I would like pit out: a) why are calf weights described to two significant figures? What is the significance of 0.14kg of a calf weighing 34kgs: it is almost meaningless, a number derived from dividing total birth weight of calves by 20! b) why is average humidity similarly described? You never use the data.c) so much in this paper is about calves/cattle in general, not focused on dairy calves. The productivity of a dairy herd does not depend solely on calf management: herd variability includes culling policy, breed, lactation characteristics, management of milking, veterinary input and herd health, farmer attitude. Focus on the advantages/disadvantages of TMR reducing the variability within your model d) it would be far better to describe all the farm-based experiment in one paragraph: as it is, this is scattered throughout the Materials and Methods section. e) carrying out visual body condition scoring (BCS) is not acceptable for any animal at any stage of its life: you have to lay hands on an animal. In any case, you never refer to BCS data in the paper, so why include it? f) you must write a scientific experiment in the past tense. In this paper, the present and past are both used throughout. g) please, do not define a phrase using brackets: this destroys the ease of reading, for examples ll 90, 99, 100, 113, 125, 126. All these qualifying phrases should be in the M & Ms as text.
Introduction
This paper is about TMR, dairy calves and your regression model. This section should describe what is known already about TMR fed to calves, its significance regarding rumen development, impact on calf health and growth, and composition. Then, why use regression analysis and how this could help farmers fedd calves better: this justifies why you have investigated TMR in dairy calves, and what your model could predict in practical terms.
M & Ms
First, describe the farm, the experimental set up, preventive medicine strategies used to maintain good calf health. Then the calves themselves: numbers male/female, housing, cleanliness, feed management with actual amounts given in kgs: weighing, definitions of time postpartum, health monitoring. Then data collection: feed, clinical and laboratory samples - amount of fees, placed where, labelled, stored, laboratory receiving them. Regarding laboratory methods, if routine, state just that: if special, describe and reference. Data storage and method used for model: define the model, the variables. Table 1 is it necessary to use two signifiant figures: this does not alter the first sig.fig of any component. Table 2 should be 2a) Ingredients 2b) chemical composition of TMR - if this paper is to be useful to famers, should you go to two significant figures - the variation is at 5% maximum so does it matter? Your call! I do not understand why 1% salt is included separately yet 20% salt is in the supplement
Results
It is up to the Editor of the journal, but it would be easier to read this paper if the Results were separate from the Discussion. If each calf received the same amount/weight of milk, there is no further comment needed: all calves remained in good health. If older/heavier calves were fed more, that should be recorded. Then TMR: Table 1 records the data so comment only on the timing of significant change around day 25, when milk was withdrawn , impact on fecal scoring Table 5 makes on sense because the subscript describes three letters al p<0.05 which do not appear related to any numerals in the table - in any case, in Table 4 you should state the differences in significant probabilities a, b, c. Table 3 is also not correct: period D0 means nothing and should go in text: why not use histogram to show mean increase in body weight and weight gain for each of the four periods, and you do not need two significant figures for these.If you describe under ‘Item’ heading what the initials stand for in full, the definitions’ subscript becomes redundant. Table 4 you do not discuss except for fecal pH: why? Data here indicates a little of the physiological changes that take place in the rumen, reflected in the feces, and could lead to comment regarding the model you created and reduction in variable ‘noise’ - its removal improves the predictability of the model!
Discussion
Must focus on the investigation described in the paper and why your model is similar/different to others without comment regarding forages fed to other species, breeds of cattle, ages of cattle other than calves. Then the predictive value of this model, to famers and other on-farm advisors: how they use it. You may extend that to the benefit to calves that fit the model for future health and productivity in the milking herd - economic as well as welfare. Finally, what were the advantages/disadvantages of your current investigation methods, and what could be done in the future to improve the predictive value of this type of analysis on TMR feeding.
References
Need to look very critically at these, remove all that are irrelevant for example what has ref90 got to do with this paper’s title - nothing. The list could be reduced to 30 references at the most.
Reasonable
Author Response
Reviewer 2:
The title of this paper is, in simple mathematical terms, a polynomial regression equation where the dependant variable y may be either solid food intake or calf body weight, b the variable you can optimise, and x the independent variable. The authors include G. Buoniauto, and they expect the reader of this paper to go to reference 34 to find out the basic mathematics associated with a non-linear association between variables x and y, defined within that paper but relating to an entirely different age and breed of cattle than in the title of this paper. The justification for this paper is that it may help dairy farmers and their nutritionists/veterinary surgeons to manage better the nutritional requirements of calves during their first 70 days of life. Therefore, this paper should be written in a clear, simple way that describes the basic research methods used to provide the data that allowed this predictive model to be constructed, the variables defined within this model, the advantages and disadvantages of using this statistical method, and how it can be used to improve the health, welfare and future lactations of milking cattle so treated.
>>> AU: Thank you for your comment, we changed the manuscript according to your and other reviewer comments. In particular we re-formulated parts of the Introduction (LL 47-49, 67-82), Material and methods (LL 84-89, 106-111, 113-121, 158-169), Results and discussion (LL 239-250, 263-264, 296-297, 325-328, 342-361) and Conclusions (LL 368-373) section to describe more clearly our methods and expand on the applicability of the research. All modifications are highlighted in yellow in the manuscript.
This objective has been lost completely in this paper: instead, it has become a review article of predictive models relating feed/rations with future performance/production of animals consuming grass/forages. Furthermore, 11% of the references come from the authors of this paper, and 18% of all references quoted are unrelated to the title of this paper. What this paper should be concerned with is the composition of the total mixed ration(TMR) fed to the dairy calves in this study, the overall nutritional management in the first 70 days of life, how this ensured that the regression equation derived was so consistent, and what that tells us about the physiological changes that occur in the rumen and digestive tract of calves within this crucial period of life which are primarily genet-related and can be influenced by external factors. These factors are important when relating your result to that of Silva - ref. 73.
>>> AU: Please see the answer above. We have reformulated entire parts of the manuscript to improve its focus and coherence/cohesion. We have also extensively reduced the reference list removing approximately 1/3 of the references. Furthermore, following your suggestion, we have made revisions to the Results and Discussion sections.
There are several other general issues that I would like pit out: a) why are calf weights described to two significant figures? What is the significance of 0.14kg of a calf weighing 34kgs: it is almost meaningless, a number derived from dividing total birth weight of calves by 20!
>>> AU: Thank you for your question. Following your comment below, we have represented the weight of the animals using a histogram.
- b) why is average humidity similarly described? You never use the data. c) so much in this paper is about calves/cattle in general, not focused on dairy calves. The productivity of a dairy herd does not depend solely on calf management: herd variability includes culling policy, breed, lactation characteristics, management of milking, veterinary input and herd health, farmer attitude. Focus on the advantages/disadvantages of TMR reducing the variability within your model d) it would be far better to describe all the farm-based experiment in one paragraph: as it is, this is scattered throughout the Materials and Methods section.
>>> AU: Thank you for your comment, Below, we address the points you raised:
- b) Regarding the mention of climatic characteristics of the area, we now understand your concern about the lack of relevance of this data within the context of the article. We acknowledge that such information might not be directly relevant to the main objectives of the article, and as a result, we have decided to remove it to maintain focus on our analysis of the advantages and disadvantages of utilizing the Total Mixed Ration (TMR) in calf management.
- c) we are aware that the mentioned factors also have a role on the health and future productivity of the dairy cows and we decided to make this observation more explicit in the manuscript (lines 348-352).
- d) Regarding the structure of the Materials and Methods section, we understand your point about the fragmentation of information related to on-farm experiments. We have followed your suggestions as much as possible and have worked to restructure and enhance the Materials and Methods sections accordingly (most information of the farm is now reported in Section 2.1).
- e) carrying out visual body condition scoring (BCS) is not acceptable for any animal at any stage of its life: you have to lay hands on an animal. In any case, you never refer to BCS data in the paper, so why include it?
>>> AU: Thank you for your valuable comment. We indeed acknowledge the importance of hands-on animal assessment for accurate results. However, it's worth noting that various researchers have utilized and published scientific works using visual BCS assessment as a reliable methodology. This approach, when performed by trained experts, has shown its efficacy in numerous studies (as example 10.3168/jds.S0022-0302(94)77212-X, 10.3168/jds.S0022-0302(97)75917-4, 10.3168/jds.2009-2431, 10.3390/ani12050601, 10.3390/ani13101668, 10.3168/jds.2022-22394)
Furthermore, we would like to emphasize that the BCS evaluations were carried out by a breed expert who possesses a high level of specialization in the field of morpho-functional assessment of cattle. This information was added to the manuscript (lines 110-111). This expert is actively engaged in BCS evaluation for two prominent cattle breeding organizations in Italy: The Italian Holstein, Italian Brown and Jersey (ANAFIBJ), and the Italian Simmental (ANAPRI). The BCS evaluation used in this study therefore, despite being only visual, meets the criteria of accuracy and validity due to the officially recognized training of the assessor.
- f) you must write a scientific experiment in the past tense. In this paper, the present and past are both used throughout.
>>> AU: we corrected the verb tenses according to the suggestion
- g) please, do not define a phrase using brackets: this destroys the ease of reading, for examples ll 90, 99, 100, 113, 125, 126. All these qualifying phrases should be in the M & Ms as text.
>>> AU: we corrected rephrased the sentences according to your suggestions (e.g., lines 85-89, 108-110 and 118-121).
Introduction
This paper is about TMR, dairy calves and your regression model. This section should describe what is known already about TMR fed to calves, its significance regarding rumen development, impact on calf health and growth, and composition. Then, why use regression analysis and how this could help farmers fedd calves better: this justifies why you have investigated TMR in dairy calves, and what your model could predict in practical terms.
>>> AU: Thank you, we reformulated the introduction to include your suggestions (lines 67-82)
M & Ms
First, describe the farm, the experimental set up, preventive medicine strategies used to maintain good calf health. Then the calves themselves: numbers male/female, housing, cleanliness, feed management with actual amounts given in kgs: weighing, definitions of time postpartum, health monitoring. Then data collection: feed, clinical and laboratory samples - amount of fees, placed where, labelled, stored, laboratory receiving them. Regarding laboratory methods, if routine, state just that: if special, describe and reference. Data storage and method used for model: define the model, the variables.
>>> AU: Thank you, we reformulated and reorganized the M&M section to include the missing information you suggested (lines 107-110, 113-121, and 158-169).
Table 1 is it necessary to use two signifiant figures: this does not alter the first sig.fig of any component.
>>> AU: modified according to the Reviewer’s suggestion.
Table 2 should be 2a) Ingredients 2b) chemical composition of TMR - if this paper is to be useful to famers, should you go to two significant figures - the variation is at 5% maximum so does it matter? Your call!
>>> AU: we apologize, but it is unclear to us what the Reviewer means by this suggestion. If they refer to Tables 1.1 and 1.2, the tables have now been unified.
I do not understand why 1% salt is included separately yet 20% salt is in the supplement
>>> AU: the salt is included both in the min&vit supplement and in the TMR
Results
It is up to the Editor of the journal, but it would be easier to read this paper if the Results were separate from the Discussion.
>>> AU: In agreement with the Journal guidelines that leave the choice to the Authors and with the lack of comments from the other reviewers, we decided to keep the two sections together to avoid redundancies in the text.
If each calf received the same amount/weight of milk, there is no further comment needed: all calves remained in good health. If older/heavier calves were fed more, that should be recorded.
>>> AU: Thank you for your comment, as stated in LL176-178, no differences were found among the calves which all finished their meals without waste.
Then TMR: Table 1 records the data so comment only on the timing of significant change around day 25, when milk was withdrawn, impact on fecal scoring.
>>> AU: Thank you for your comment, done as suggested.
Table 5 makes on sense because the subscript describes three letters al p<0.05 which do not appear related to any numerals in the table - in any case, in Table 4 you should state the differences in significant probabilities a, b, c.
>>> AU: Thank you for your suggestion, we modified tables 3 and 4 according to your suggestions.
Table 3 is also not correct: period D0 means nothing and should go in text: why not use histogram to show mean increase in body weight and weight gain for each of the four periods, and you do not need two significant figures for these. If you describe under ‘Item’ heading what the initials stand for in full, the definitions’ subscript becomes redundant.
>>> AU: Thank you for your suggestion, we modified table 2 according to your suggestions and create figure 2 (LL 263-265).
Table 4 you do not discuss except for fecal pH: why? Data here indicates a little of the physiological changes that take place in the rumen, reflected in the feces, and could lead to comment regarding the model you created and reduction in variable ‘noise’ - its removal improves the predictability of the model!
>>> AU: We present the results of the fecal output in lines 234-246. Regarding the discussion, we were unable to find studies that reported compositional characteristics of feces from healthy calves. The studies we came across primarily focused on subjects with intestinal issues, making it seem inappropriate to compare such data with ours. However, if the reviewer is aware of studies reporting such data, we would be more than willing to discuss the obtained results.
Discussion
Must focus on the investigation described in the paper and why your model is similar/different to others without comment regarding forages fed to other species, breeds of cattle, ages of cattle other than calves. Then the predictive value of this model, to famers and other on-farm advisors: how they use it. You may extend that to the benefit to calves that fit the model for future health and productivity in the milking herd - economic as well as welfare. Finally, what were the advantages/disadvantages of your current investigation methods, and what could be done in the future to improve the predictive value of this type of analysis on TMR feeding.
>>> AU: Thank you, we shaped a part of the discussion following your suggestions (lines 293-294, 322-325, and 339-358).
References
Need to look very critically at these, remove all that are irrelevant for example what has ref90 got to do with this paper’s title - nothing. The list could be reduced to 30 references at the most.
>>> AU: Thank you for your comment. The reference list has been thoroughly reconsidered and 1/3 of the reference was removed. We removed the citations that were less pertinent to the aim of the research. We believe that all the references that are now cited are now pertinent and contribute to supporting and contextualizing the concepts presented in the text. In our revision, we took into account the fact that the journal’s guidelines do not impose any restrictions on the number of citations to be included. and the positive feedback from other reviewers who have deemed our manuscript to be well-written. We are open to further discussion on this matter and are willing to consider any additional feedback you may provide.
Reviewer 3 Report
Overall, the pape is novel and well written. I have a few comments for improvement.
In the title, I think there should be an "a" after "calves fed"
Line 43: What are some of the minimum standards?
Line 76: I am not sure why you say "preliminary" here. I would remove that wording because it seems like when you say that the study is not conclusive at all. I would say that you developed a an equation to predict intake in calves.
Line 88: Why was 77 days chosen?
Line 89-90: How practical is it to take BCS scores on calves? They change so much and BCS in calves would have a very different meaning than BCS in lactating cows.
Table 1: I am not sure this is needed in a table. Either list this in Table 2 or just include it in the text.
Line 124: How often were calves weighed? Weekly? daily?
Line 153: This statistical analysis section is lacking a lot of information. I am not sure how the stat analysis was conducted. Please be specific in the model effects for the mixed models, as well as the polynomial regression model. You don't really describe very well on how you were going to predict intake in calves. How?
Table 6: You don't really explain much about your prediction equation. Why day 39? It is hard to quantify the values in the equation without knowing how you did it in the stat analysis? More detail is neded.
Line 345: I don't think you need these limitations in your study. It does diminish the results that you got. I would remove this paragraph.
Author Response
Reviewer 3
Overall, the pape is novel and well written. I have a few comments for improvement.
>>> AU: Thank you for appreciating our study.
In the title, I think there should be an "a" after "calves fed"
>>> AU: Thank you for your suggestion. We have modified the title.
Line 43: What are some of the minimum standards?
>>> AU: Thank you for your question. Please note that this part of the manuscript was removed in agreement with the suggestion from Reviewer 2 to improve the focus of the manuscript. The Council Directive 2008/119/EC includes minimum standards regarding rearing spaces ("The width of any individual pen for a calf shall be at least equal to the height of the calf at the withers, measured in the standing position, and the length shall be at least equal to the body length of the calf, measured from the tip of the nose to the caudal edge of the tuber ischii (pin bone), multiplied by 1.1."). Therefore, we do not have a precise measurement (as is the case for group rearing of calves). In our case, the pens measured 2.2 x 1.5 m, as detailed in the "Materials and Methods," L 90. Additional requirements are given for the provision of feed (L97) and water (LL 98-99), bedding (L 91), social contact (L90) and mutilations. The information useful for the description of the research are now reported in the Materials and methods section. All structures used for the experiment followed the directive. Additionally, we have obtained ethical committee approval to extend the individual cage rearing period by 16 days for research purposes (LL 87-89).
Line 76: I am not sure why you say "preliminary" here. I would remove that wording because it seems like when you say that the study is not conclusive at all. I would say that you developed a an equation to predict intake in calves.
>>> AU: We sincerely appreciate your thorough review of our scientific manuscript and your suggestion regarding the use of the term "preliminary." The decision to use this adjective was made with utmost transparency, reflecting our awareness that, despite successfully developing a model, it remains in a phase of ongoing improvement and optimization. The term "preliminary" was chosen to convey that while we have achieved promising results, our model needs to be improved by integrating of data from other farms to enrich and diversify our dataset and obtain a higher degree of generalizability. In agreement with the reviewer’s suggestion, the word “preliminary” was removed from the introduction but left in other parts of the manuscript
Line 88: Why was 77 days chosen?
>>> AU: Thank you for your question. 77 days were chosen in order to observe the complete transition of the calves during the weaning period (line 114)
Line 89-90: How practical is it to take BCS scores on calves? They change so much and BCS in calves would have a very different meaning than BCS in lactating cows.
>>> AU: Thank you for your question. As indicated in M&Ms section (LL 89-91), we decided to assess the BCS as a complementary parameter to verify the proper health status of the calves. This step ensured that we included only healthy calves in our study. We utilized this parameter, along with other factors, as a screening method. The BCS assessment was carried out by a breed expert trained for this kind of evaluation also in young animals. Given that our research involved very young animals, we aimed to minimize stress by avoiding excessive blood sampling and other invasive procedures. Our focus was on confirming the suitability of their body conformation, especially considering the age of the animals involved in our study.
Table 1: I am not sure this is needed in a table. Either list this in Table 2 or just include it in the text.
>>> AU: Thank you for your comment. We modified table 2 according to the suggestion of another reviewer.
Line 124: How often were calves weighed? Weekly? daily?
>>> AU: We chose to weigh the calves at 5 specific ages (please see lines 113-116). We opted for such a low frequency of weighing because, as we were housing the calves in individual cages, the weighing procedures could potentially be stressful for the animals.
Line 153: This statistical analysis section is lacking a lot of information. I am not sure how the stat analysis was conducted. Please be specific in the model effects for the mixed models, as well as the polynomial regression model. You don't really describe very well on how you were going to predict intake in calves. How?
>>> AU: Thank you, we added information on the equation in the m&m section.
Table 6: You don't really explain much about your prediction equation. Why day 39? It is hard to quantify the values in the equation without knowing how you did it in the stat analysis? More detail is neded.
>>> AU: Thank you, we added information on the equation in the m&m section.
Line 345: I don't think you need these limitations in your study. It does diminish the results that you got. I would remove this paragraph.
>>> AU: Thank you for your comment. This section was left (although a bit sofrened) in agreement with the concerns expressed by another reviewer.
Round 2
Reviewer 3 Report
The authors have satisfactorily responded to my comments.